## LETTER TO THE EDITOR

# Evolutionary poker lacks a full deck when modelling the LTEE Cit+ phenotype

The recent paper by Root-Bernstein & Bernstein (2024) in *The Journal of Physiology* on the E-poker modelling of Lenski's Long-Term Evolution Experiment (LTEE) cites our work (van Hofwegen et al., 2016). We appreciate the insightful exploration by the authors, especially regarding the role of interactomes and evolutionary processes. We would like to clarify several points that refer to our findings and add context to the general discussion about evolutionary contingency and interpreting the LTEE *Escherichia coli* Cit+ phenotype.

### Clarification of Gould's contingency theory

Root-Bernstein and Bernstein (p. 2523) cite that we explicitly conclude that the LTEE negates Gould's theory of contingency. To clarify, we are referencing Lenski's early interpretation (Pennesi, 2013). Lenski initially thought all LTEE *E. coli* populations followed a uniform evolutionary path but changed his view after a Cit+ variant emerged in one of 12 flasks after 15 years. Indeed, after another 20 years, additional Cit+ variants have not arisen in the LTEE giving the impression that the route to this phenotype is contingent on rare antecedent mutations and not repeatable. Using basic genetic analysis, our results show Cit+ mutations are not due to contingency. Gould's theory may be true, just not in this circumstance. The key points of our results are the reversible mechanism by which aerobic citrate transport arises and the genetic differences in succinate metabolism between *E. coli* K12 and *E. coli* B.

### Aerobic citrate mutations have a reversible intermediate phenotype

Root-Bernstein and Bernstein (p. 2523) state, '….no previous strain of *E. coli* had ever been observed to be able to catabolize citrate in aerobic conditions'. To clarify, Lenski states in his original LTEE Cit+ paper (Blount et al., 2008) that aerobic citrate use by *E. coli* had only been observed once before (Hall, 1982) but gives no details. Hall's paper describes the isolation of *E. coli* K12 Cit+ variants in 14 days compared

to the LTEE 15 years. We simply asked, 'what could account for this significant time difference between these two experiments?'

Hall's work was done when genomic sequencing was not available to genotype his mutation. He mapped his mutation to the region of the chromosome responsible for anaerobic citrate metabolism and speculated he had activated the anaerobic citrate transporter. To pick up on his work, we first repeated his experiments. We isolated multiple independent Cit+ mutants that arose between 12 and 100 generations using a direct selection with citrate as the sole carbon source (a direct selection in contrast to the LTEE as Root-Bernstein and Bernstein rightly identify). Importantly, genomic sequencing shows the type/class of our Cit+ mutants are analogous to Lenski's lone Cit+ LTEE isolate. They all arise from an amplification in the citrate transporter gene, *citT*, followed by a promoter capture by genome restructuring/fusion with an aerobically expressed nearby gene. Gene amplifications are the most common mutations in *E. coli* and importantly *they are reversible* and subject to intermittent selection, for and against, during the 24 h cycle of the LTEE protocol. We verified this point by repeating Lenski's protocol but transferred cultures after 1 week instead of 24 h. After one or two transfers (1–2 weeks), Cit+ variants arise in multiple independent experiments. This is because extended time in stationary phase with citrate allows cells with *citT* amplifications to increase in number and thus increase the probability of a promoter capture. Roth and Maisnier-Patin (2016) agree with our interpretation that intermittent selection was at play *vs.* contingency.

Finally, using phage transduction, we show that the Cit+ mutation can be transferred into wild-type *E. coli* K12 without prior citrate exposure, genetic proof that the aerobic Cit+ phenotypes do not require contingent single nucleotide polymorphism accumulation.

### *E. coli* B *versus* *E. coli* K12: genetic differences in succinate metabolism impact Cit+ adaptation

Root-Bernstein and Bernstein rightly point out (their fig. 9) that two transporters are required for aerobic citrate metabolism, CitT (citrate antiporter) and

DctA (C4-dicarboxylate transporter). The authors further state, 'Mutations in both transporters are necessary in order for aerobic citrate use to evolve and become adaptive'. To clarify, a mutation is required for aerobic *citT* expression (as explained above), but a mutation to express *dctA is only required in E. coli B*, the strain Lenski used in the LTEE. CitT is an antiporter; it transports citrate into the cell as it expels a C4-dicarboxylic acid (e.g. succinate). Six carbons in, four carbons out. For the full oxidative potential of growth with citrate, the expelled C4-dicarboxylate must be brought back into the cell by the DctA transporter. This transporter of C4-dicarboxylates *is under the control of the dcuSR two-component regulatory system*. For *E. coli* K12, when succinate is in the environment, it binds to the membrane-bound sensor DcuS, a histidine kinase, which auto-phosphorylates, and then transfers the phosphate to DcuR (response regulator) and the DcuR~P binds to and activates the *dctA* promoter to express DctA. Hence, *E. coli* K12 can grow on succinate as a sole carbon source and needs no mutation to import succinate. This is not true for *E. coli* B. In this strain, DctA is not expressed due to a *dcuS* 5 base pair deletion/frameshift mutation, evidenced by the fact that wild-type *E. coli* B cannot grow on succinate. It took 33,000 generations for *E. coli* B to acquire both aerobic CitT expression and a suppressor of the *dcuS* mutation to become fully Cit+. This suppressor is a single base pair mutation in the *dctA* promoter overriding the need for the *dcuSR* system. Figures 1 and 2 illustrate these key differences between *E. coli* B and *E. coli* K12.

Lenski and others argue that our results using both a direct selection in citrate medium and delayed transference of cultures in low glucose + citrate medium are irrelevant regarding acquisition of the Cit+ phenotype in the LTEE (e.g. see discussion of Leon et al., 2018, pp. 12–13). We argue otherwise. In fact, what is remarkable in both sets of experiments with *E. coli* K12 (direct selection or a short, delayed selection with glucose) and the *E. coli* B LTEE conditions is that each produced the same result requiring aerobic expression of CitT and DctA. Even Darwin, as reviewed by Roth (2012), speculated that a strong *vs.* weak selection would

The Journal of Physiology

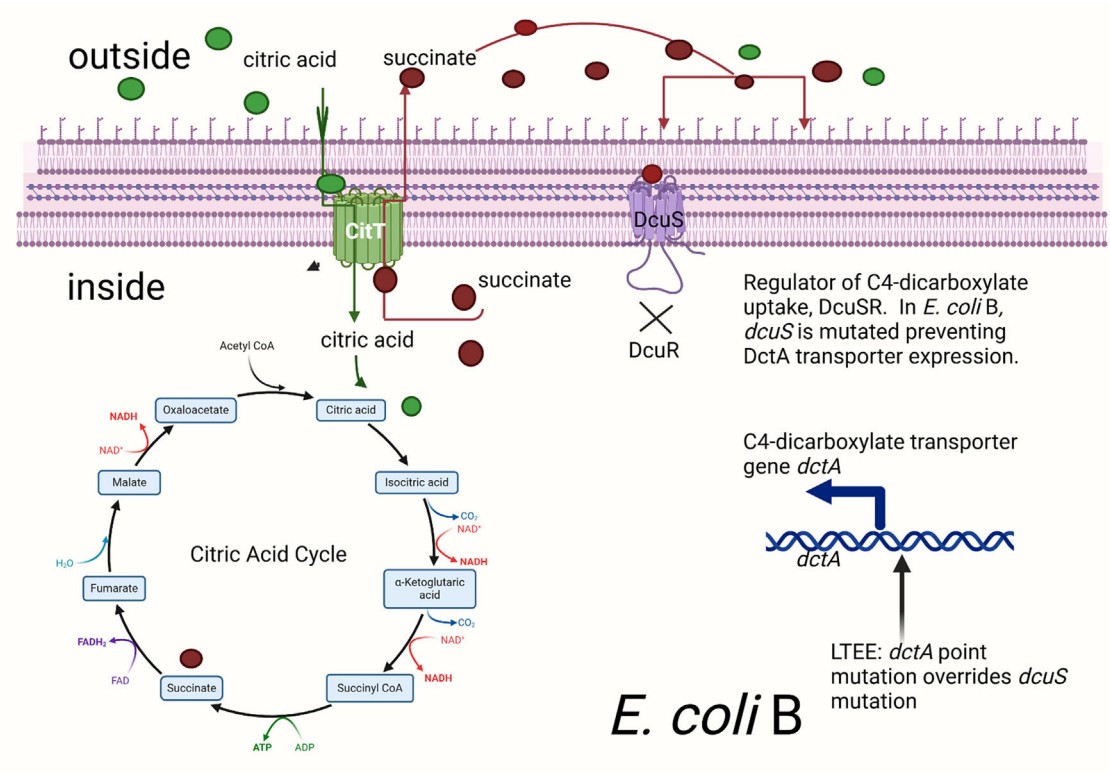

**Figure 1. *E. coli* B aerobic citrate utilization requires constitutive expression of CitT and a suppressor mutation of *dcuS⁻***

The CitT antiporter (green complex in the inner membrane) is expressed aerobically after *citT* amplification followed by chromosome remodelling to capture an aerobic promoter. The antiporter function of CitT is depicted as uptake of external citrate (green spheres) and export of a C4-dicarboxylic acid, such as succinate (red spheres). At this initial stage, import of citrate yields a net gain of two carbons for metabolism. However, as growth continues and succinate accumulates outside the cell, succinate cannot be recaptured. This is because in *E. coli* B DcuS is defective and cannot phosphorylate DcuR (see X under DcuS) preventing expression of DctA that is required for C4-dicarboxylate uptake. Overcoming the lack of the DcuSR two-component system requires a point mutation of the *dctA* promoter. This took 15 years (33,000 generations) to occur in LTEE and only happened in one of the 12 populations.

produce different outcomes. Our results combined with the LTEE results show that whether the selection is strong or weak, the outcomes are phenotypically and genotypically equivalent for aerobic citrate use by *E. coli*. We think this is perhaps the most significant finding of these combined experimental approaches. Furthermore, the combined probability of getting both aerobic CitT expression and the *dctA* point mutation suppressing the need of *dcuS* is $\sim 1 \times 10^{-14}$. This figure is derived from: (i) gene amplifications are the most common mutations in bacteria (Roth, 2012) and we estimate that getting aerobic expression of CiT via gene amplification/recombination in the order of $\sim 1 \times 10^{-5}$; and (ii) a point mutation to suppress *dcuS⁻* is in the range of $1 \times 10^{-9}$ to $10^{-10}$. The fact that the LTEE Cit⁺ mutation was acquired after 33,000 generations ($\sim 1 \times 10^{14}$ cells, see Pennesi, 2013) likewise argues against contingency.

In conclusion, Lenski asserts that the LTEE's most important outcome is the rare and innovative Cit⁺ mutation. Had he chosen *E. coli* K12, rather than *E. coli* B (Lenski's starting strain which lacks a functional key gene in its genetic deck), it is likely that all 12 flasks would rapidly have become Cit⁺ and we would not be arguing about contingency and the requirement of rare and necessary non-repeatable antecedent mutations.

Scott A. Minnich and Carolyn J. Hovde

*Animal, Veterinary, and Food Science,
University of Idaho, Moscow, Idaho, USA*

Email: sminnich@uidaho.edu

Handling Editors: Laura Bennet & Michael Joyner

The peer review history is available in the Supporting Information section of this article (https://doi.org/10.1113/JP288355#support-information-section).

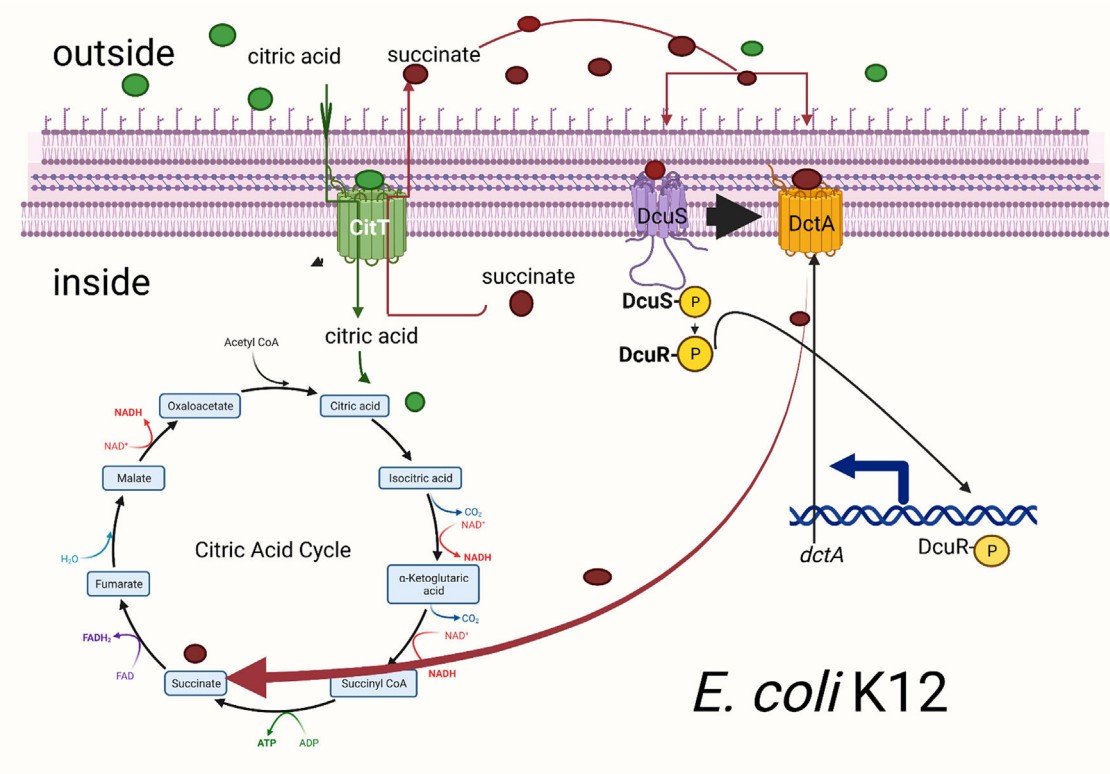

**Figure 2.** *E. coli* **K12 aerobic citrate utilization only requires constitutive expression of CitT.**
The CitT antiporter (green complex in the inner membrane) is expressed aerobically after *citT* amplification followed by chromosome remodelling to capture an aerobic promoter. The antiporter function of CitT is depicted as uptake of external citrate (green spheres) by CitT driven by export of a C4-dicarboxylic acid, such as succinate (red spheres). As growth continues and succinate accumulates in the environment, it binds to the DcuS sensor protein of the DcuSR two-component system (purple complex in inner membrane). This binding induces DcuS to auto-phosphorylate (DcuS~P) which then phosphorylates DcuR. DcuR~P transcriptionally activates *dctA*, coding for a C4-dicarboxylate transporter. The DctA protein transporter, orange complex in the inner membrane, recaptures the four carbons lost by the CitT antiporter. Imported succinate can feed directly into the citric acid cycle and be fully oxidized to $CO_2$ and $H_2O$.

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

## Additional information

### Competing interests

No competing interests declared.

## Author contributions

Scott Minnich: Conception or design of the work; Drafting the work or revising it critically for important intellectual content; Final approval of the version to be published; Agreement to be accountable for all aspects of the work. Carolyn Hovde: Conception or design of the work; Drafting the work or revising it critically for important intellectual content; Final approval of the version to be published; Agreement to be accountable for all aspects of the work. All authors have approved the final version of the manuscript and agree to be accountable for all aspects of the work. All persons designated as authors qualify for authorship, and all those who qualify for authorship are listed.

## Funding

This publication was made possible by an Institutional Development Award (IDeA) from

the National Institutes of Health under Grant #P20GM103408.

## Acknowledgements

The figures were created with Biorender.com.

## Keywords

contingency, escherichia coli Cit$^+$, evolution modeling, long-term evolution experiment

## Supporting information

Additional supporting information can be found online in the Supporting Information section at the end of the HTML view of the article. Supporting information files available:

**Peer Review History**

