## [Peer Review History · The Journal of Physiology]

Evolutionary Poker lacks a full deck when modeling the LTEE Cit+ phenotype

Scott A. Minnich and Carolyn J. Hovde
DOI: 10.1113/JP288355

Corresponding author(s): Scott Minnich (sminnich@uidaho.edu)

The following individual(s) involved in review of this submission have agreed to reveal their identity: Michael J. Joyner (Referee #1)

Review Timeline:

Submission Date:	12-Dec-2024
Editorial Decision:	14-Jan-2025
Revision Received:	20-Jan-2025
Accepted:	22-Jan-2025

Senior Editor: Laura Bennet

Reviewing Editor: Michael Joyner

Transaction Report:

Dear Dr Hovde,

Re: JP-LE-2024-288355 "Evolutionary Poker lacks a full deck when modeling the LTEE Cit+ phenotype" by Scott A. Minnich and Carolyn J. Hovde

Thank you for submitting your manuscript to The Journal of Physiology. It has been assessed by a Reviewing Editor and by 2 expert referees and we are pleased to tell you that it is acceptable for publication following satisfactory revision.

Please address all the points raised and incorporate all requested revisions or explain in your Response to Referees why a change has not been made. We hope you will find the comments helpful and that you will be able to return your revised manuscript within 4weeks. If you require longer than this, please contact journal staff: jp@physoc.org.

REVISION CHECKLIST:

- 'Potential Cover Art' for consideration as the issue's cover image
- Appropriate Supporting Information (Video, audio or data set: see https://jp.msubmit.net/cgi-bin/main.plex?form_type=display_requirements#supporting_information)

form_type=display_requirements#supp).

We look forward to receiving your revised submission.

Yours sincerely,

Laura Bennet
Senior Editor
The Journal of Physiology

REQUIRED ITEMS

- Please include a full title page as part of your main article (Word) file, which should contain the following: title, authors, affiliations, corresponding author name and contact details, keywords, and running title.

EDITOR COMMENTS

Reviewing Editor:

The comments of reviewer 2 need attention before final acceptance.

Please also see 'Required Items' above.

REFEREE COMMENTS

Referee #1:

This is an excellent brief letter that makes some key points and shares the authors perspective on a key issue in experimental evolution. This is the sort of dialogue that can add context and move a field forward especially in the context of interpreting iconic foundational experiments.

Referee #2:

This is a useful follow-up and clarification to the paper by Root-Bernstein and Bernstein.

The subject matter is technical enough that I cannot judge its accuracy.

Some of the references cited in the manuscript lack years. Please fix.

Line 47: Add "their" before referencing Fig. 9

Line 64: Give some specific references here for "Lenski and others ..."

Line 68-69: Reference for Darwin? The Origin? Edition? Pages?

Line 72-73: Where does this probability value come from? Explain the calculation, even if it's simple.

Line 78: It seems like a final sentence should be added, saying something like "These results also demonstrate how the starting (base) population for a selection experiment can have a major effect on the response to selection by constraining the initial genetic and phenotypic toolbox (references)." References might be something like these, but you'd need to check to make sure they make this point:

Bell, G. 2013. Responses to selection: experimental populations. Pp. 232-239 in D. A. Baum, ed. *The Princeton Guide to Evolution*. Princeton University Press.

Brennan, R. S., J. A. deMayo, H. G. Dam, M. Finiguerra, H. Baumann, V. Buffalo, and M. H. Pespeni. 2022. Experimental evolution reveals the synergistic genomic mechanisms of adaptation to ocean warming and acidification in a marine copepod. *Proceedings of the National Academy of Sciences* 119:e2201521119.

Garland, Jr., T., and M. R. Rose (eds). 2009. *Experimental evolution: concepts, methods, and applications of selection experiments*. University of California Press, Berkeley.

Kawecki, T. J., R. E. Lenski, D. Ebert, B. Hollis, I. Olivieri, and M. C. Whitlock. 2012. Experimental evolution. *Trends Ecol. Evol. (Amst.)* 27:547-560.

Sanders, B. R., L. S. Thomas, N. M. Lewis, Z. A. Ferguson, J. L. Graves, and M. D. Thomas. 2024. It takes two to make a thing go right: epistasis, two-component response systems, and bacterial adaptation. *Microorganisms* 12:2000.

Line 117-118: Title should not be capitalized

END OF COMMENTS

January 20, 2015.

Dear Laura Bennet, Senior Editor:

Thank you for considering the publication of our letter. Below are our responses to the referee's comments in green. We believe their insights and our resulting edits have improved the clarity of our writing for your readership.

Sincerely,
Scott A. Minnich and Carolyn J. Hovde

Referee #1:

This is an excellent brief letter that makes some key points and shares the authors perspective on a key issue in experimental evolution. This is the sort of dialogue that can add context and move a field forward especially in the context of interpreting iconic foundational experiments.

Thank you for these comments. We made no changes.

Referee #2:

This is a useful follow-up and clarification to the paper by Root-Bernstein and Bernstein.

The subject matter is technical enough that I cannot judge its accuracy.

Some of the references cited in the manuscript lack years. Please fix.

Done. We amended the citations in the text to include the year.

Line 47: Add "their" before referencing Fig. 9

Done. "their" has been added to the text

Line 64: Give some specific references here for "Lenski and others ..."

Done. We added the Leon et al. (2018) reference as an example. This paper states '*that isolating Cit⁺ mutants by direct selection are irrelevant to the LTEE results*'. Additional criticisms of our paper by Lenski were from a blog, which we have not referenced.

Line 68-69: Reference for Darwin? The Origin? Edition? Pages?

Agree. Adding a reference is important. Rather than cite The Origin directly (Darwin's Chapter 4 in *Origins* focuses on "Natural Selection") we referenced a review by Roth (2012) from Microbes and Evolution. This short review deals specifically with the strength of selection. Also, this review addresses the role of gene duplications, and their frequency during phenotypic changes. We think readers will find this review most beneficial regarding strong versus weak selective pressure because Roth addresses a similar case to Cit⁺ in *E. coli*, the restoration of the Lac⁺ phenotype in phenotypic *E. coli* Lac⁻ cells.

Line 72-73: Where does this probability value come from? Explain the calculation, even if it's simple.

Agree. We added the probabilities of the two required mutations, individually.

Line 78: It seems like a final sentence should be added, saying something like "These results also demonstrate how the starting (base) population for a selection experiment can have a major effect on the response to selection by constraining the initial genetic and phenotypic toolbox (references)." References might be something like these, but you'd need to check to make sure they make this point:

Agree and appreciate this comment. The strain of *E. coli* one starts with is the point of our letter. To address the reviewer concern, we changed the last sentence to: "*Had he chosen E. coli K12, rather than E. coli B (which lacks a functional key gene in its starting genetic deck), it is likely that all 12 flasks would rapidly have become Cit⁺ and we would not be arguing about contingency and the requirement of rare and necessary non-repeatable antecedent mutations.*" We reviewed the reviewer's suggested references and feel they are either too far afield or are not necessary.

Line 117-118: Title should not be capitalized

Agree. Good catch, citation has been corrected.

Dear Dr Minnich,

Re: JP-LE-2025-288355R1 "Evolutionary Poker lacks a full deck when modeling the LTEE Cit+ phenotype" by Scott A. Minnich and Carolyn J. Hovde

We are pleased to tell you that your paper has been accepted for publication in The Journal of Physiology.

Yours sincerely,

Laura Bennet
Senior Editor
The Journal of Physiology

If you would like to receive our 'Research Roundup', a monthly newsletter highlighting the cutting-edge research published in The Physiological Society's family of journals (The Journal of Physiology, Experimental Physiology, Physiological Reports, The Journal of Nutritional Physiology, and The Journal of Precision Medicine: Health and Disease), please click this link, fill in your name and email address and select 'Research Roundup':

<https://www.physoc.org/journals-and-media/membernews>

- You can help your research get the attention it deserves! Check out Wiley's free Promotion Guide for best-practice recommendations for promoting your work at: www.wileyauthors.com/eoo/guide. You can learn more about Wiley Editing Services which offers professional video, design, and writing services to create shareable video abstracts, infographics, conference posters, lay summaries, and research news stories for your research at: www.wileyauthors.com/eoo/promotion.

The Corresponding Author will receive an email from Wiley with details on how to register or log-in to Wiley Authors Services where you will be able to place an order

EDITOR COMMENTS

Reviewing Editor:

No further edits needed.